# Evaluation of the Association between Low-Density Lipoprotein (LDL) and All-Cause Mortality in Geriatric Patients with Hip Fractures: A Prospective Cohort Study of 339 Patients

**DOI:** 10.3390/jpm13020345

**Published:** 2023-02-16

**Authors:** Xin Kang, Bin Tian, Zan-Dong Zhao, Bin-Fei Zhang, Ming Zhang

**Affiliations:** 1Department of Sport Medicine, Honghui Hospital, Xi’an Jiaotong University, No. 555 Youyi East Road, Xi’an 710054, China; 2Department of Joint Surgery, Honghui Hospital, Xi’an Jiaotong University, No. 555 Youyi East Road, Xi’an 710054, China; 3Department of General Medicine, Honghui Hospital, Xi’an Jiaotong University, No. 555 Youyi East Road, Xi’an 710054, China

**Keywords:** LDL, mortality, hip fractures, cohort study

## Abstract

Background: Many factors affect the prognosis of hip fractures in the elderly. Some studies have suggested a direct or indirect association among serum lipid levels, osteoporosis, and hip fracture risk. LDL levels were found to have a statistically significant nonlinear U-shaped relationship with hip fracture risk. However, the relationship between serum LDL levels and the prognosis of patients with hip fractures remains unclear. Therefore, in this study, we assessed the influence of serum LDL levels on patient mortality over a long-term follow-up period. Methods: Elderly patients with hip fractures were screened between January 2015 and September 2019, and their demographic and clinical characteristics were collected. Linear and nonlinear multivariate Cox regression models were used to identify the association between LDL levels and mortality. Analyses were performed using Empower Stats and R software. Results: Overall, 339 patients with a mean follow-up period of 34.17 months were included in this study. Ninety-nine patients (29.20%) died due to all-cause mortality. Linear multivariate Cox regression models showed that LDL levels were associated with mortality (HR = 0.69, 95%CI: 0.53, 0.91, *p =* 0.0085) after adjusting for confounding factors. However, the linear association was unstable, and nonlinearity was identified. An LDL concentration of 2.31 mmol/L was defined as the inflection point for prediction. A LDL level < 2.31 mmol/L was associated with mortality (HR = 0.42, 95%CI: 0.25, 0.69, *p =* 0.0006), whereas LDL > 2.31 mmol/L was not a risk factor for mortality (HR = 1.06, 95%CI: 0.70, 1.63, *p* = 0.7722). Conclusions: The preoperative LDL level was nonlinearly associated with mortality in elderly patients with hip fractures, and the LDL level was a risk indicator of mortality. Furthermore, 2.31 mmol/L could be considered a predictor cut-off for risk.

## 1. Introduction

Osteoporosis is characterized by reduced bone mass and strength, which increases the risk of fragility fractures [1,2,3], and causes long-term severe pain and/or dysfunction, seriously affecting patients’ quality of life [4,5,6,7]. Osteoporosis and osteoporotic fractures become more common with advancing age. Worldwide, osteoporotic fractures accounted for 0.83% of the global burden of non-communicable diseases, increasing to 1.75% of the burden in Europe [8]. Total fragility fractures in the EU are estimated to increase by 23%, from 2.7 million in 2017 to 3.3 million in 2030, and the resulting annual fracture-related costs (EUR 37.5 billion in 2017) are expected to increase by 27%. An estimated 1.0 million quality-adjusted life years (QALYs) are lost due to these fractures, and the disability-adjusted life years (DALYs) are higher than the estimates for stroke, chronic obstructive pulmonary disease, and common cancers, with the exception of lung cancer.^2,8^ Osteoporotic fractures typically occur in the hip, spine, wrist, and humerus. Fractures of the hip are among the most common and serious sites of osteoporotic fracture, which account for the majority of fracture-related healthcare expenditures and mortalities in men and women over the age of 50 years [8,9,10,11,12,13,14]. This poses a heavy burden on both individuals and society due to high treatment costs, reduced health-related quality of life, and reduced survival [15]. Fracture-related burdens are expected to continue increasing in the coming decades [2]. Therefore, preventive identification and prompt intervention for the risk of geriatric hip fractures are needed in these patients.

Many factors affect the prognosis of hip fractures in the elderly population. Pneumonia and circulatory system diseases are the most common causes of death in this population, and the mortality risk factors with a higher relative risk are advanced age, male sex, increased comorbidities, delirium, and medical complications during admission. Underlying risk factors include decompensation of chronic illness, fracture-related functional decline, and malnutrition. Patients with worse conditions at admission also have the highest risk of mortality [16,17,18,19].

Low-density lipoproteins (LDL), termed “bad cholesterol,” are large molecules comprising many proteins and lipids, including cholesterol, phospholipids, and triglycerides. Oxidized low-density lipoproteins (Ox-LDL) modulate the innate and adaptive immune responses, and can act in both pro- and anti-inflammatory manners through many proposed mechanisms [20,21,22,23]. Some studies have suggested a direct or indirect association among serum lipid levels, osteoporosis, and hip fracture risk [24,25,26,27]. In a prospective cohort study following 5832 participants aged ≥ 65 years from the Cardiovascular Health Study for hip fracture for a mean of 13.5 (SD 5.7) years. LDL levels were found to have a statistically significant nonlinear U-shaped relationship with hip fracture risk (*p* = 0.02) [28]. LDL cholesterol comprises 90% of the circulating cholesterol in most people; therefore, there is a high correlation between total cholesterol and LDL levels [29].

However, the relationship between serum LDL levels and the prognosis of patients with hip fractures remains unclear. Therefore, in this study, we assessed the influence of serum LDL levels on patient mortality over a long-term follow-up period. We hypothesized that there would be either a linear or nonlinear association between LDL levels and mortality. This prospective cohort study aimed to identify the role of LDL levels in hip fractures.

## 2. Materials and Methods

### 2.1. Study Design

We recruited elderly patients who were treated for hip fractures between 1 January 2015 and 30 September 2019 at the largest trauma center in Northwest China. This prospective study was approved by the Ethics Committee of the Xi’an Honghui Hospital (No. 202201009). All procedures involving human participants were performed in accordance with the 1964 Declaration of Helsinki and its amendments.

### 2.2. Participants

The demographic and clinical data of the patients were obtained from their original medical records. The inclusion criteria were as follows: (1) age ≥ 65 years; (2) a radiographic or computed tomography diagnosis of a femoral neck, intertrochanteric, or subtrochanteric fracture; (3) patients receiving surgical or conservative treatment in a hospital; (4) availability of clinical data in the hospital; and (5) patients able to be contacted by telephone. Patients who could not be contacted were excluded from the study.

### 2.3. Hospital Treatment

The patients were examined using blood tests and ultrasonography to prepare for surgery. Intertrochanteric fractures are often managed with closed/open reductions and internal fixations of the proximal femoral nail by antirotation. Femoral neck fractures are often treated with hemiarthroplasty or total hip arthroplasty, depending on the patient’s age. Prophylaxis for deep vein thrombosis was initiated on admission. Upon discharge, the patients were asked to return for monthly check-ups to assess fracture union or function.

### 2.4. Follow-Up

After discharge, the patients’ family members were contacted by telephone from January 2022 to March 2022 to record data on survival, survival time, and activities of daily living. This follow-up was conducted by two medical professionals with two weeks of training and one year of experience. Contact was attempted two more times for patients who could not be contacted initially. If the family members could not be contacted, we recorded the patient as lost to follow-up.

### 2.5. Endpoint Events

The endpoint event in this study was all-cause mortality after treatment. We defined all-cause mortality as death reported by patients’ family members.

### 2.6. Variables

The variables in our study were as follows: age, sex, occupation, history of allergy, injury mechanism, fracture classification, presence of hypertension, diabetes, coronary heart disease, arrhythmia, hemorrhagic stroke, ischemic stroke, cancer, associated injuries, dementia, chronic obstructive pulmonary disease (COPD), hepatitis and gastritis, time from injury to admission, time from admission to operation, LDL level, duration of surgery, blood loss, infusion, transfusion, treatment, total hospital stay, and follow-up.

LDL level was defined as the liver function in the blood test performed at admission. If a patient did not undergo surgery for any reason, the final results before discharge were selected. The dependent variable was all-cause mortality, while the independent variable was LDL level. The other variables were defined as potentially confounding factors.

### 2.7. Statistical Analysis

Continuous variables are reported as the mean ± standard deviation (Gaussian distribution) or median (range, skewed distribution). Categorical variables are presented as numbers with proportions. Chi-square (categorical variables), one-way analysis of variance (ANOVA (normal distribution)), or Kruskal–Wallis H test (skewed distribution) were performed to detect the differences in different LDL levels. Univariate and multivariate Cox proportional hazard regression models (three models) were used to test the association between LDL levels and mortality. Model 1 was not adjusted for covariates. Model 2 was minimally adjusted only for sociodemographic variables. Model 3 was fully adjusted for all covariates. To test the robustness of our results, we performed a sensitivity analysis. We converted the LDL level into a categorical variable according to the anemia criteria, calculated *p* for the trend to verify the results of LDL as a continuous variable, and examined the possibility of nonlinearity. Because Cox proportional hazards regression model-based methods are suspected to be unable to deal with nonlinear models, the nonlinearity between LDL and mortality was addressed using a Cox proportional hazard regression model with cubic spline functions and smooth curve fitting, termed the penalized spline method. If nonlinearity was detected, we first calculated the inflection point using a recursive algorithm and then constructed a two-piecewise Cox proportional hazards regression model on both sides of the inflection point.

All analyses were performed using statistical software packages R (http://www.R-project.org, R Foundation for Statistical Computing, Vienna, Austria) and EmpowerStats (http://www.empowerstats.com, X&Y Solutions Inc., Boston, MA, USA). Hazard ratios (HR) and 95% CI were calculated. Statistical significance was set at *p* < 0.05 (two-sided).

## 3. Results

### 3.1. Patient Characteristics

Overall, 399 patients treated between January 2015 and September 2019 were included in this study. The mean follow-up period was 34.17 months, and 99 patients (29.20%) died due to all-cause mortality. LDL concentrations were divided into three groups. Table 1 lists the demographic and clinical characteristics of the 399 patients, including comorbidities, factors associated with injuries, and treatment.

### 3.2. Univariate Analysis of Association between Variates and Mortality

We performed univariate analysis to identify potential confounding factors and the relationship between variables and mortality (Appendix A). According to the criteria of *p* < 0.1, the following variables were considered in the multivariate Cox regression: age, CHD, arrhythmia, dementia, and treatment strategy.

### 3.3. Multivariate Analysis between LDL and Mortality

We used three models (Table 2) to correlate LDL levels and mortality. When LDL concentration was a continuous variable, linear regression was observed. The fully adjusted model showed a decrease in mortality risk (HR = 0.69, 95%CI: 0.53–0.91, *p =* 0.0085) when LDL concentration increased by 1 mmol/L after controlling for confounding factors. When LDL concentration was used as a categorical variable, we found statistically significant differences in LDL levels among the three models (*p* < 0.0001). In addition, the *p* for trend also showed a linear correlation in the three models (*p* < 0.0001).

However, we found that the changing interval was slow in the subgroups with different LDL levels (Table 2). This instability indicates the possibility of a nonlinear correlation.

### 3.4. Curve Fitting and Analysis of Threshold Effect

As shown in Figure 1, there was a curved association between LDL levels and mortality after adjusting for confounding factors. We compared two fitting models to explain this association (Table 3). Interestingly, we observed an inflection point in the saturation effect at 2.31 mmol/L. This indicates that at LDL < 2.31 mmol/L, the mortality risk decreased by 58% (HR = 0.42, 95%CI: 0.25–0.69; *p* = 0.0006) when LDL concentration increased by 1 mmol/L; when LDL > 2.31 mmol/L, the mortality risk did not decrease with a LDL change (HR = 1.06, 95%CI: 0.70–1.63; *p* = 0.7722).

The Kaplan–Meier survival curves according to LDL level (*p* < 0.0001) and the inflection point of 2.31 mmol/L (*p* = 0.0016) are shown in Figure 2.

## 4. Discussion

In this study, we identified a nonlinear association between LDL and all-cause mortality in geriatric hip fractures, finding that when LDL < 2.31 mmol/L, the mortality risk decreased by 58% with an LDL concentration increase of 1 mmol/L (HR = 0.42, 95%CI: 0.25–0.69; *p* = 0.0006); conversely, when LDL > 2.31 mmol/L, the mortality risk did not decrease with LDL change (HR = 1.06, 95%CI: 0.70–1.63; *p* = 0.7722). LDL < 2.31 mmol/L could be considered a predictor of the risk of increased mortality in clinical settings, with a lower LDL level being associated with higher mortality. The LDL results were unexpected, indicating that the lowest levels of LDL were associated with the highest risk of mortality following hip fractures.

At present, most related studies have focused on the association between lipid levels and osteoporosis risk, finding conflicting results. Some studies have suggested positive associations, some report no associations, and others report negative associations [30,31,32,33,34]. At the same time, studies on hip fractures are limited. Although a follow-up study showed that lipids and lipoproteins are associated with hip fracture risk in older adults, no relationship between LDL levels and the prognosis or all-cause mortality in geriatric hip fractures has been identified. Therefore, in this study, we explored the relationship between LDL levels and the prognosis of hip fractures in the elderly to provide further evidence of the relationship between LDL and geriatric hip fractures.

In addition to the linear relationship, we speculatively identified the existence of a curvilinear relationship through subgroup analysis and curve fitting. We were further able to find an inflection point in the curve. For this reason, the curve linear relationship is more appropriate to explain the relationship between LDL levels and geriatric hip fracture mortality.

A prior cohort study showed that the association between LDL levels and the risk of all-cause mortality was U-shaped, with low and high levels of LDL being associated with an increased risk of all-cause mortality. An LDL concentration of 3.6 mmol/L indicated the lowest risk of all-cause mortality. The association between low levels of LDL and an increased risk of all-cause mortality could be explained by reverse causation [35]. Debilitation and illness could decrease cholesterol levels, especially in elderly hospitalized patients, and comorbidities were more frequent in individuals with the lowest levels of LDL [36,37]. A survival analysis in China showed that a lower-admission LDL level (LDL < 2.755 mmol/L) was associated with an increased risk of long-term mortality in acute aortic dissection (HR = 3.287, 95%CI: 1.637–6.600, *p* = 0.001) [38].

The “cholesterol paradox” could also explain our results. This paradox states that low cholesterol is related to a worse prognosis and higher mortality. Several studies on cardiovascular diseases support this conclusion. For example, some studies on heart failure and acute myocardial infarction have shown that a lower baseline LDL increases the risk of patient mortality [39,40,41,42]. Physiologically, LDL is critical for the synthesis of cellular membranes and steroid hormones. Several factors may account for the “cholesterol paradox,” including a higher proportion of elderly patients, a higher proportion of baseline comorbidities, and malnutrition [43,44,45]. Some previous studies have shown a significantly negative association between LDL and bone mineral density (BMD), thereby increasing fracture incidence and all-cause mortality [46,47,48]. A cohort study of bone mineral density and 5-year mortality in end-stage renal disease patients previously showed that low total BMD were independent predictors of increased risk of all-cause mortality [49]. The same conclusion was drawn in several studies of hemodialysis patients [50,51]. Furthermore, low LDL levels have also been reported to be associated with decreased cognitive function, depression, and mood disorders, which could affect prognosis [52].

The strengths of our study include the following: First, as a prospective cohort study, we tried our best to avoid a loss to follow-up. Patients who could not be contacted were excluded from the study. Second, information on the cause of death for each individual was reported by the patients’ family members. Third, we adjusted for several confounders with an effect on mortality risk as well as LDL levels [53,54,55,56,57] to control for the majority of confounding factors.

However, this study has some limitations. First, loss to follow-up is unavoidable in a prospective cohort study, and this study is no exception. Therefore, we performed multiple telephone follow ups with those patients who could not be contacted initially to obtain patients’ outcome information. Second, this study was not able to determine the causal relationship between LDL levels and geriatric hip fracture prognosis; this will need to be confirmed in future studies. Third, our study population was derived only from western China; therefore, the conclusions may have geographical and ethnic limitations. Caution should be exercised when using this conclusion for other population groups.

In summary, we found that the preoperative LDL level was nonlinearly associated with mortality in elderly hip fracture patients, and a low LDL level was a risk indicator of mortality. Furthermore, an LDL concentration of 2.31 mmol/L could be considered a predictor cut-off for risk.

## Figures and Tables

**Figure 1 jpm-13-00345-f001:**
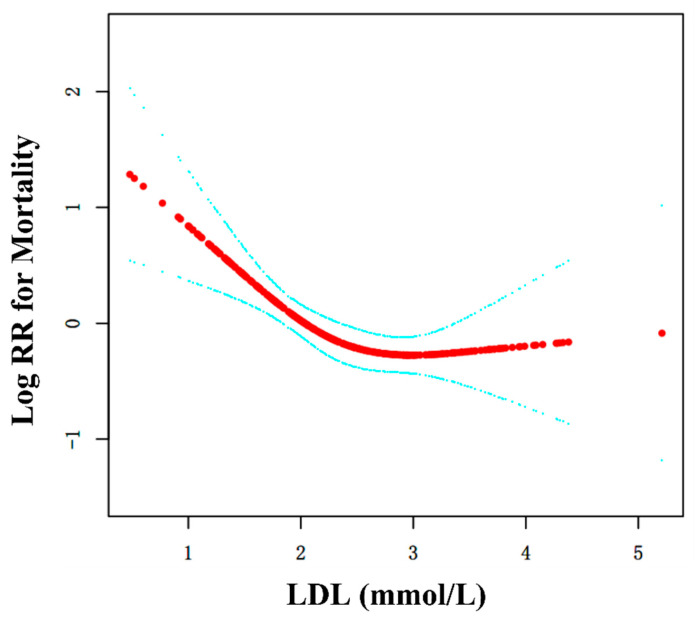
Curve fitting between LDL and mortality. Adjusted for age, sex, CHD, arrhythmia, dementia, treatment strategy. The red line is the fitting curve, and the blue lines are 95%CI.

**Figure 2 jpm-13-00345-f002:**
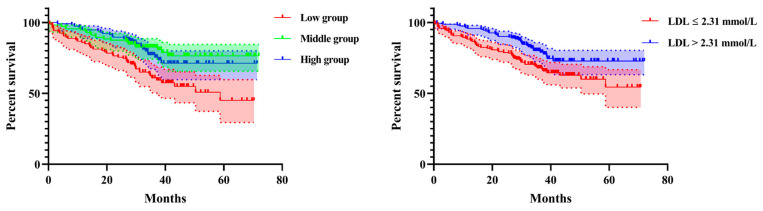
The Kaplan–Meier survival curve according to LDL levels and inflection point of 2.31 mmol/L.

**Table 1 jpm-13-00345-t001:** Demographic and clinical characteristics.

LDL Tertiles	Low	Middle	High	*p*-Value	*p*-Value *
N	111	112	116		
LDL	1.54 ± 0.33	2.29 ± 0.19	3.27 ± 0.48	<0.001	<0.001
Age (years)	81.36 ± 6.30	79.76 ± 6.45	79.01 ± 6.45	0.02	0.032
Sex				0.028	-
Male	45 (40.54%)	39 (34.82%)	28 (24.14%)		
Female	66 (59.46%)	73 (65.18%)	88 (75.86%)		
Occupation				0.618	-
Retirement	62 (55.86%)	61 (54.46%)	68 (58.62%)		
Farmer	29 (26.13%)	23 (20.54%)	23 (19.83%)		
Other	20 (18.02%)	28 (25.00%)	25 (21.55%)		
History of allergy	11 (9.91%)	7 (6.25%)	9 (7.76%)	0.598	-
Injury mechanism				0.427	0.379
Falling	106 (95.50%)	109 (97.32%)	115 (99.14%)		
Accident	3 (2.70%)	1 (0.89%)	1 (0.86%)		
Other	2 (1.80%)	2 (1.79%)	0 (0.00%)		
Fracture classification				0.009	0.005
Intertrochanteric fracture	87 (78.38%)	83 (74.11%)	69 (59.48%)		
Femoral neck fracture	24 (21.62%)	25 (22.32%)	43 (37.07%)		
Subtrochanteric fracture	0 (0.00%)	4 (3.57%)	4 (3.45%)		
Hypertension	60 (54.05%)	64 (57.14%)	72 (62.07%)	0.466	-
Diabetes	24 (21.62%)	19 (16.96%)	27 (23.28%)	0.477	-
CHD	57 (51.35%)	53 (47.32%)	56 (48.28%)	0.82	-
Arrhythmia	44 (39.64%)	29 (25.89%)	25 (21.55%)	0.008	-
Hemorrhagic stroke	1 (0.90%)	3 (2.68%)	5 (4.31%)	0.279	0.311
Ischemic stroke	44 (39.64%)	39 (34.82%)	37 (31.90%)	0.47	-
Cancer	2 (1.80%)	4 (3.57%)	3 (2.59%)	0.712	0.777
Multiple injuries	4 (3.60%)	9 (8.04%)	7 (6.03%)	0.372	-
Dementia	6 (5.41%)	7 (6.25%)	6 (5.17%)	0.934	-
COPD	6 (5.41%)	7 (6.25%)	5 (4.31%)	0.807	-
Hepatitis	2 (1.80%)	2 (1.79%)	2 (1.72%)	0.999	1
Gastritis	1 (0.90%)	3 (2.68%)	3 (2.59%)	0.575	0.707
Treatment strategy				0.002	-
Conservation	14 (12.61%)	2 (1.79%)	7 (6.03%)		
ORIF	73 (65.77%)	84 (75.00%)	65 (56.03%)		
HA	24 (21.62%)	25 (22.32%)	43 (37.07%)		
THA	0 (0.00%)	1 (0.89%)	1 (0.86%)		
Time to admission (h)	128.73 ± 530.08	100.21 ± 203.31	94.80 ± 263.95	0.75	0.819
Time to operation (d)	4.61 ± 2.04	4.63 ± 2.24	4.79 ± 2.60	0.827	0.985
Operation time (mins)	101.02 ± 41.34	96.36 ± 32.08	100.28 ± 38.61	0.621	0.975
Blood loss (mL)	247.66 ± 156.49	212.67 ± 114.02	241.40 ± 159.01	0.183	0.441
Infusion (mL)	1521.08 ± 354.91	1587.92 ± 360.94	1597.77 ± 383.82	0.289	0.259
Transfusion (U)	1.24 ± 1.27	1.12 ± 1.22	0.87 ± 1.26	0.1	0.055
Length in hospital (d)	8.59 ± 3.46	8.35 ± 2.63	8.45 ± 2.67	0.82	0.954
Follow-up (months)	31.02 ± 17.91	36.62 ± 15.02	34.86 ± 13.09	0.023	0.039
Mortality	49 (44.14%)	22 (19.64%)	28 (24.14%)	<0.001	-

Mean + SD/N(%)*. p*-value *: For continuous variables, we used the Kruskal–Wallis rank-sum test and Fisher’s exact probability test for count variables with a theoretical number of <10.

**Table 2 jpm-13-00345-t002:** Univariate and multivariate results by cox regression analyses.

Exposure	Non-Adjusted Model	Minimally-Adjusted Model	Fully Adjusted Model
LDL	0.61 (0.46, 0.80) 0.0005	0.67 (0.50, 0.88) 0.0048	0.69 (0.53, 0.91) 0.0085
LDL tertiles			
Low	Ref	Ref	Ref
Middle	0.38 (0.23, 0.63) 0.0002	0.42 (0.25, 0.69) 0.0007	0.48 (0.29, 0.81) 0.0058
High	0.48 (0.30, 0.77) 0.0022	0.57 (0.35, 0.91) 0.0192	0.61 (0.37, 0.99) 0.0434
*p* for trend	0.0012	0.01	0.0263

Data in table: HR (95%CI) *p*-value Outcome variable: mortality Exposed variables: LDL, Minimally adjusted model adjusted for: age; sex. Fully adjusted model adjusted for age, sex, CHD, arrhythmia, dementia, and treatment strategy.

**Table 3 jpm-13-00345-t003:** Nonlinearity of LDL and mortality.

Outcome	HR (95%CI), *p*-Value
Fitting model by stand linear regression	0.69 (0.53, 0.91), 0.0085
Fitting model by two-piecewise linear regression	
Inflection point	2.31 mmol/L
<2.31 mmol/L	0.42 (0.25, 0.69), 0.0006
>2.31 mmol/L	1.06 (0.70, 1.63), 0.7722
*p* for log-likelihood ratio test	0.024

Adjust for: age, sex, CHD, arrhythmia, dementia, treatment strategy.

## Data Availability

The data were provided by Xi’an Honghui Hospital. According to relevant regulations, the data cannot be shared, but could request from correspondence author.

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
