# Peer review of "Evaluation of the Association between Low-Density Lipoprotein (LDL) and All-Cause Mortality in Geriatric Patients with Hip Fractures: A Prospective Cohort Study of 339 Patients"

_jpm, 2023, doi:10.3390/jpm13020345_

Round 1

Reviewer 1 Report

The paper by Kang et al. addresses a very interesting subject, one of the associations between low-density lipoprotein and all-cause mortality in the elderly population with hip fractures. The manuscript is well written, and the review protocols and data scientifically sound and are presented clearly. Nonetheless, some minor modifications could improve their work quality.

After reading the manuscript the following issues raised my concern or represent suggestions that in my opinion could increase the quality of the manuscript:

-       Lines 85-87 in my opinion should be merged, because the sentence from line 85 doesn’t have a meaning.

-       “Patients who could not be contacted were excluded, and were asked to return monthly to assess fracture union or function upon discharge.” Maybe should be rewritten because if you do not pay attention, you might understand that the patients that couldn’t be contacted were asked to return…

-       Please reassess the manuscript for tying mistakes. 

Author Response

Comments and Suggestions for Authors The paper by Kang et al. addresses a very interesting subject, one of the associations between low-density lipoprotein and all-cause mortality in the elderly population with hip fractures. The manuscript is well written, and the review protocols and data scientifically sound and are presented clearly. Nonetheless, some minor modifications could improve their work quality. After reading the manuscript the following issues raised my concern or represent suggestions that in my opinion could increase the quality of the manuscript: Lines 85-87 in my opinion should be merged, because the sentence from line 85 doesn’t have a meaning. Response: Thanks for your suggestion. We have deleted the sentence,”Lipids and lipoproteins are also associated with hip fracture risk in older adults.” “Patients who could not be contacted were excluded, and were asked to return monthly to assess fracture union or function upon discharge.” Maybe should be rewritten because if you do not pay attention, you might understand that the patients that couldn’t be contacted were asked to return… Response: Thanks for your suggestion. We have changed the sentence. Please check. Please reassess the manuscript for tying mistakes. Response: Thanks for your suggestion. We have rechecked the words. Please check.

Reviewer 2 Report

Thank you for the opportunity to review the interesting manuscript titled: Evaluation of the Association between Low Density Lipoprotein (LDL) and All-Cause Mortality in Geriatric Patients with Hip Fractures: A Prospective Cohort Study of 339 Patients

The article is very well written, I consider that the article can be accepted as long as the following is addressed:

- More biological support is needed for the LDL paradox, it is not enough to describe previous studies. It is necessary to substantiate this paradox biologically and molecularly. More biological support is needed for the results of this study.

- Add confidence intervals in kaplan meier plots

Author Response

Response to Reviewer 2 Comments

Comments and Suggestions for Authors

Thank you for the opportunity to review the interesting manuscript titled: Evaluation of the Association between Low Density Lipoprotein (LDL) and All-Cause Mortality in Geriatric Patients with Hip Fractures: A Prospective Cohort Study of 339 Patients

The article is very well written, I consider that the article can be accepted as long as the following is addressed:

- More biological support is needed for the LDL paradox, it is not enough to describe previous studies. It is necessary to substantiate this paradox biologically and molecularly. More biological support is needed for the results of this study.

Response: Thanks for your suggestion. More biological support is needed for the results of this study. We have rewritten one paragraph to support this association in lines 245-259. Please review.

- Add confidence intervals in kaplan meier plots

Response: Thanks for your suggestion. We have changed the Kaplan-Meier plots, please check.

Round 2

Reviewer 1 Report

Dear authors,

The performed changes improved the quality of the manuscript. In my opinion the manuscript is suitable to be published. 

Reviewer 2 Report

The authors of this work kindly attended to my comments. I consider that the article can be accepted for publication.